# Determination of the Polyphenol Composition of Raspberry Leaf Using LC-MS/MS

**DOI:** 10.3390/molecules30040970

**Published:** 2025-02-19

**Authors:** Hind Mesfer S. Alkhudaydi, Esther Njeri Muriuki, Jeremy P. E. Spencer

**Affiliations:** 1Department of Food and Nutritional Sciences, School of Chemistry, Food and Pharmacy, University of Reading, Reading RG6 6DZ, UK; 2Food Science and Nutrition Department, Faculty of Science, Taif University, Taif 26571, Saudi Arabia

**Keywords:** polyphenolic profile, raspberry leaf, LC-MS, principal component analysis

## Abstract

Background: Raspberry leaf (RL; *Rubus idaeus*) is a by-product of raspberry cultivation and has been proposed to be a rich source of micronutrients and potential bioactive components, including polyphenols. However, the precise chemical composition of the non-nutrient (poly)phenols in RL has not been as extensively studied. Objective: To evaluate the (poly)phenolic content of six RL samples from different geographical locations and to explore the impact of brewing duration on the levels of phenolic compounds available for absorption following consumption. Methods: A total of 52 polyphenolic constituents were investigated in the RL samples using Liquid Chromatography–Mass Spectrometry (LC-MS), and RL tea samples were analysed for ellagitannins, flavonoids, and phenolic acids. Tea samples were extracted using 80:20 (*v*/*v*) methanol/acidified water (0.1% formic acid) to maximise polyphenol recovery, with two sonication steps (30 and 25 min), followed by centrifugation, filtration, and storage at −18 °C. Extractions were performed in triplicate for comprehensive profiling. Additionally, raspberry leaf tea (2 g) was brewed in 200 mL of boiling water at various times (0.5–20 min) to simulate standard consumption practices; this was also performed in triplicate. This approach aimed to quantify polyphenols in the brew and identify optimal steeping times for maximum polyphenol release. Results: Raspberry leaf (RL) samples from six geographical sources were analysed, with 37 compounds identified in methanol and 37 in water out of the 52 targeted compounds, with only 7 compounds not detected in either methanol or water extracts. The analysis indicated that the total measured polyphenol content across the six samples from various sources ranged between 358.66 and 601.65 mg/100 g (*p* < 0.001). Ellagitannins were identified as the predominant polyphenolic compound in all RL samples, ranging from 155.27 to 394.22 mg/100 g. The phenolic acid and flavonoid concentrations in these samples exhibited a relatively narrow range, with the phenolic acids spanning from 38.87 to 119.03 mg/100 g and the flavonoids ranging from 125.03 to 156.73 mg/100 g. When brewing the tea, the 5 min extraction time was observed to yield the highest level of polyphenols (505.65 mg/100 g) (*p*< 0.001), which was significantly higher than that with shorter (409.84 mg/100g) and longer extraction times (429.28 mg/100 g). Notably, ellagic acid levels were highest at 5 min (380.29 mg/100 g), while phenolic acid peaked at 15 min (50.96 mg/100 g). The flavonoid content was shown to be highest at 4 min (82.58 mg/100 g). Conclusions: RL contains a relatively high level of polyphenols, particularly ellagic acid; thus, its consumption may contribute to the daily intake of these health-beneficial non-nutrient components.

## 1. Introduction

Polyphenols are naturally occurring compounds found in a variety of plant-based foods and beverages, such as dark chocolate, tea, coffee, fruits, and vegetables [1,2]. They have been extensively investigated for their potential to contribute to health by reducing inflammation, improving vascular function, regulating blood sugar levels [3,4], and promoting cognitive health [5,6]. These compounds are widely accessible through the diet, especially from fruits and vegetables, such as berries, apples, olives, and green tea, and their regular consumption is thought to contribute to reducing chronic disease risk [7,8]. For example, (−)-epicatechin, a flavanol found in tea, cocoa, and apple, may induce acute, dose-dependent improvements in vascular function, with effects evident at doses as low as 0.5 mg/kg body weight [9]. In addition to flavonoids, dietary supplementation with phenolic-rich olive leaf extract has been shown to significantly lower blood pressure, total cholesterol, LDL cholesterol, triglyceride, and interleukin-8 levels in pre-hypertensive males, highlighting its potential cardiovascular benefits [10].

Raspberry (*Rubus* spp.) is a genus in the Rosaceae family and is primarily cultivated in temperate zones in the northern hemisphere [11], with cultivation originating in Europe [12]. Raspberry leaf (RL) is considered a by-product of raspberry production, similar to olive leaf [13], and like the latter, both its leaves and its fruit are thought to be high in bioactive components, and it is included in the British Pharmacopoeia [14]. RL is discarded in large quantities yearly [15,16,17] and is significantly underutilised compared to raspberry fruit [18]. Currently, most RL harvested as a by-product of raspberry production is used as fertiliser [19,20], even though previous reports suggest that it may contain a diverse array of bioactive substances [21], including polyphenols [22,23,24,25,26,27]. For example, refs. [28,29] identified gallic acid, ellagic acid, and procyanidin as significant components of RL extract, and other studies have identified vitamins (B1, B2, B3, B6, E, C, A, and K) [30,31,32,33]; organic acids such as citric and malic acids; minerals; and dietary fibre [31,32,33]. This rich composition of nutrients and non-nutrients has raised the possibility that the consumption of RL extracts may be capable of inducing health benefits, in a similar manner to that observed for other polyphenol-rich foods and beverages such as tea, vegetables, and fruits [7,33].

Previous data suggest that RL may induce anti-inflammatory, chemo-preventive, and antibacterial properties [34], and its regular consumption may induce antidiabetic, anti-obesity, anti-inflammatory, hepatoprotective, and immunomodulatory properties [35,36,37,38,39,40,41,42,43]. Additionally, RL has been shown to impact body weight control in mice [44,45]. Indeed, both the berries and the leaves/young shoots of RL have a long history of use in traditional folk medicine to treat common colds, fever, gastrointestinal ailments, and menstrual cramps, and to stimulate labour [35,39,46,47]. Although RL is widely utilised in traditional medicine to aid pregnancy and labour, the available evidence on its safety and effectiveness is inconclusive [48]. While some research indicates potential benefits, such as a shorter labour duration [49,50,51], other studies highlight concerns regarding its uterotonic properties and the absence of rigorous clinical trials to confirm its safety [48,52,53]. However, it can be used to treat gastrointestinal disorders, respiratory issues, heart problems, influenza, fever, and diabetes [54]. Data have indicated that RL’s primary bioactive compounds closely resemble those found in the raspberry fruit [21,39]. However, the precise composition of the bioactive compounds within RL and its impact on human cardiometabolic health have not been fully defined.

Despite previous research identifying polyphenols in *Rubus idaeus* (RL), existing knowledge remains limited and lacks comprehensive analysis. This study aims to extensively evaluate the full spectrum of (poly)phenolic compounds in RL, examining variations in polyphenol content across different RL tea products and under varying extraction conditions. By exploring these variations, we seek to determine whether RL tea constitutes a significant dietary source of polyphenols.

## 2. Results

### LC-MS/MS Analysis of Polyphenol Compounds in Raspberry Leaf Samples

In this study, LC-MS/MS was used to analyse (poly)phenols in RL samples sourced from Belgium, Croatia, the UK, Bulgaria, Germany, and Poland to ensure geographical diversity. LC-MS analysis of water/methanol RL tea extractions was used to characterise and quantify 37 of the 52 standard compounds tested (Table 1), including flavanones, phenolic acids, and ellagic acid. Significant variations in ellagitannin, phenolic acid, and flavonoid content were observed across RL tea samples (A–F) (*p* < 0.001). Tea samples C (601.65 mg/100 g) and F (601.38 mg/100 g) exhibited the highest polyphenol content, significantly exceeding other samples (*p* < 0.001), while tea A had the lowest content (358.66 mg/100 g). For ellagitannins, tea F had the highest levels (394.22 mg/100 g), followed by tea C, with tea D showing the lowest concentration (155.27 mg/100 g). The phenolic acid content was the highest in tea B (119.03 mg/100 g) and the lowest in tea A (38.87 mg/100 g). Similarly, the flavonoid levels peaked in tea B (156.73 mg/100 g), while tea F had the lowest flavonoid content (125.03 mg/100 g). Detailed data are presented in Table 1 and Figure 1, illustrating these compositional differences and their significance. These results highlight the significant geographical and compositional variation in RL teas.

Water-only extractions of tea C were conducted to simulate conventional brewing practices. We assumed that all samples would respond similarly in the brewing experiment. Therefore, we selected the one with the highest total content for our analysis. This led to the identification of a similar profile of compounds to that detected in the water/methanol extractions, although some components, such as m-coumaric acid, diosmin, and kaempferol, were either absent or present in lower concentrations (Figure 2; Table 2). Brewing time influenced the concentration of polyphenols, flavonoids, phenolic acids, and ellagitannins in raspberry leaf tea, with the total levels peaking at 505.65 mg/100 g after 5 min of hot water extraction (Figure 2). The flavonoid concentration reached its highest level of 82.58 mg/100 g at 4 min; this dropped to 69.89 mg/100 g by 20 min. Meanwhile, phenolic acids showed minor fluctuations, with the highest concentration of 50.96 mg/100 g observed at 15 min and the lowest of 38.90 mg/100 g at 0.5 min. Ellagitannin content, specifically ellagic acid, showed a similar trend, peaking at 380.29 mg/100 g at 5 min and declining to 311.95 mg/100 g at 20 min (Figure 2; Table 2).

## 3. Discussion

In this study, we employed LC-MS/MS to assess the levels of non-nutrient phytochemicals in six RL tea samples. We initially used water/methanol extraction to obtain a comprehensive chemical profile, with 37 of the 52 targeted polyphenol compounds detected in RL samples sourced from six countries (Belgium, Croatia, the UK, Bulgaria, Germany, and Poland). This geographical diversity enabled an analysis of variations in polyphenol content due to varying environmental conditions across Europe. Significant differences were observed among the samples, with the ellagitannin ellagic acid being the most abundant compound detected in all samples, with nearly two-fold differences in amount. Phenolic acids and flavonoids were also detected in all samples and expressed a similar range of variation in total levels. To simulate consumer preparation, we also extracted the teas in hot water. Both extraction methods resulted in the identification and quantification of 37 of the 52 targeted polyphenols. Extraction in hot water led to significant differences in total polyphenols levels (*p* < 0.001), with a peak at 5 min before decreasing; ellagitannins and flavonoids followed a similar trend, suggesting that there may have been some oxidation of compounds occurring at later extraction times [55,56,57]. Phenolic acids showed minor differences in total levels and peaked later at 15 min. Considering a typical 5 min brewing time, the total level of polyphenols was 505.65 mg/100 g (ellagic acid: 380.29 mg/100 g; total flavonoids: 77.56 mg/100 g; and total phenolic acids: 47.80 100 mg/g). Our data confirm that ellagic acid is the predominant component in raspberry leaf (RL), aligning with previous studies [29,58]. However, the concentration we measured (380.29 mg/100 g) falls within the range of previously reported values but is notably higher than some studies, such as 157.36 mg/100 g [29], 194.11 mg/100 g [44], 281.30 mg/100 g [58], and 292.20 mg/100 g [59], while being lower than others, such as 438 mg/100 g [60]. In contrast, significantly lower concentrations have been documented in other studies, including 5.86 mg/100 g [46] and 2.54 mg/100 g [22]. Similarly, caffeic acid concentrations exhibit considerable variability across studies, ranging from 0.52 mg/100 g [46] to 78.20 mg/100 g [60], with our research reporting an intermediate value of 12.62 mg/100 g. Specifically, higher levels of gallic acid (19.68 mg/100 g) and chlorogenic acid (104.16 mg/100 g) were detected, with the latter representing the highest concentration recorded to date compared with [29,44,46,59,60,61]. Conversely, our study found lower concentrations of quercetin (13.92 mg/100 g), kaempferol (5.54 mg/100 g), (+)-catechin (4.7 mg/100 g), p-coumaric acid (2.04 mg/100 g), and quercetin-3-rutinoside (14.94 mg/100 g) compared to the highest values reported in prior research [22,25,29,44,46,58,59,60,61,62]. We also observed notable differences in polyphenol concentrations compared to earlier findings. These results indicate variability in compound concentrations depending on extraction methods and sources.

The levels found in various RL samples and across different studies may reflect several variables, including differences in environmental conditions, extraction methods and analytical setups, and the RL variety. Factors such as cultivation, region, weather, ripeness, harvest time, and storage conditions can influence the chemical composition of polyphenols in plants [17,40]. For example, research indicates that higher altitudes and increased UV radiation enhance the production of phenolic compounds and other secondary metabolites as part of the plant’s defence response [63,64,65]. In addition, the stage of ripeness and harvest timing are critical, as phenolic compound levels may increase or decrease as the plant/fruit matures [31,66]. For example, seasonal variations also impact the phytochemical composition, with leaves harvested in different seasons exhibiting variations in flavonoid and tannin levels [40,66]. In addition, research on blackberries revealed significant fluctuations in ellagic acid levels throughout different ripening stages, with the highest concentration occurring during the green fruit stage [67]. Moreover, the timing of leaf harvest during the growing season, along with the leaf’s position (apical, middle, or basal), influences polyphenol levels, with studies showing that chlorophylls, carotenoids, total phenolic content, and total flavonoid content in RL increase steadily from May to October, and that apical leaves accumulate higher levels of specific polyphenols, such as quercetin-3-glucuronide and kaempferol-3-rutinoside compared to basal leaves [66]. Additionally, the total concentration of secondary metabolites in RL peaks in late autumn due to reduced light and temperature, which promotes significant phenolic accumulation [68,69]. However, cultivation techniques may also influence the polyphenolic composition of RL, including farming practices, pesticide and fertiliser use, and irrigation strategies [61]. For instance, a study revealed that organic RL had considerably higher levels of total polyphenols, phenolic acids, and specific compounds such as chlorogenic acid and quercetin-3-O-rutinoside [61]. Notably, organic cultivation often results in higher polyphenol concentrations due to the absence of synthetic fertilisers and pesticides, which impact plant metabolism [61]. Unlike conventional farming, which uses chemical fertilisers and pesticides to mitigate environmental stress, organic practices expose plants to natural stressors such as drought, nutrient deficiencies, and pest attacks, stimulating the production of higher levels of secondary metabolites, such as polyphenols, as part of their natural defence mechanisms [70,71,72,73,74,75]. However, variations in reported concentrations may still arise due to differences in extraction and analysis techniques [14]. Of course, the yield and composition of polyphenols are markedly influenced by extraction methodology, particularly solvent type, temperature, and duration, and techniques such as maceration, ultrasound-assisted extraction, and supercritical fluid extraction also impact total levels [76,77,78]. Research indicates that freeze-drying preserves phenolic compounds more effectively than oven-drying, which can promote oxidation and reduce bioactive compound levels [79,80,81,82,83,84].

The results presented in Figure 2 demonstrate a significant decrease in polyphenol content after 15 min of tea infusion. This finding is consistent with earlier research indicating the instability of polyphenols when exposed to hot water for extended periods [85,86]. The decrease in polyphenol concentration between 5 and 15 min could be due to various chemical and enzymatic processes, such as oxidation, polymerisation, and degradation [87,88,89]. Polyphenols, especially catechins and flavonoids, are highly prone to oxidative reactions when exposed to high temperatures and oxygen [90]. During the infusion process, polyphenol oxidation can result in the formation of quinones and other secondary products, potentially contributing to the observed decrease in polyphenol content [91]. Polyphenol polymerisation, which intensifies at higher temperatures, can also produce more significant, insoluble compounds that conventional analytical methods may fail to detect [92]. Polyphenol oxidation, catalysed by enzymatic activity such as polyphenol oxidase and intensified at elevated temperatures, can lead to the formation of quinones and secondary products, while polymerisation may produce more significant, insoluble compounds that remain undetectable by conventional analytical methods, collectively contributing to the decline in polyphenol content [85]. Furthermore, the degradation of polyphenols into smaller molecules, such as phenolic acids, along with oxidation and polymerisation, may contribute to the observed changes in polyphenol content over time [88].

The polyphenol content of RL, measured at 505 mg/100 g, highlights its potential as a rich source of these bioactive compounds. Compared to other well-known polyphenol-rich foods, RL contains notably higher concentrations. For example, black olives contain approximately 569 mg per 100 g [93], while highbush blueberries have about 560 mg per 100 g [93]. For example, oleuropein, a key polyphenol derived from tyrosol found in black olives, may positively affect blood pressure and inflammation [94,95], and the regular consumption of highbush blueberries has been linked to various health benefits, including potential improvements in cognitive function and blood pressure [96,97]. A review of current research on RL suggests that it may exhibit comparable effects in managing chronic diseases [54]. For example, RL extract has been shown to positively influence the gut microbiota during in vitro digestion and fermentation, suggesting a role in intestinal health [42]. These effects indicate that RL extract may support gut microbiome balance and contribute to obesity management [38,39,40,41,42,43]. Additionally, RL extract has demonstrated significant inhibitory effects on α-glucosidase and α-amylase enzymes in in vitro studies, thereby enhancing glucose metabolism and promoting glucose consumption in cells [98]. This evidence indicates a potential role for RL in glycaemic control, although no studies have specifically examined its impact on adult glucose regulation. Given its relatively high polyphenol content, RL is a promising candidate for dietary disease prevention and management strategies. Comparative analyses with other polyphenol-rich foods further highlight its potential efficacy in promoting human health. These findings underscore the importance of solvent optimisation, tea sample selection, and brewing practices to maximise health benefits.

## 4. Materials and Methods

Reagents and Chemicals: The standards referred to in this study, along with their corresponding Chemical Abstracts Service (CAS) numbers presented in parentheses, were purchased from Sigma Aldrich (Dorset, UK). These standards include quercetin (849061-97-8), cryptochlorogenic acid (905-99-7), benzoic acid (65-85-0), naringenin (67604-48-2), phloridizin (60-81-1), caffeic acid (331-39-5), chlorogenic acid (327-97-9), neochlorogenic acid (906-33-2), ferulic acid (537-98-4), gallic acid (149-91-7), p-Coumaric acid (501-98-4), salicylic acid (69-72-7), rosmarinic acid (20283-92-5), ellagic acid (476-66-4), kaempferol (520-18-3), myricetin (529-44-2), catechin (154-23-4), epicatechin (490-46-0), quercetin 3-O-glucoside (482-35-9), isoquercitrin (482-35-9), quercitrin (522-12-3), hyperoside (482-36-0), chlorogenic acid (327-97-9), quercetin-3-glucuronide (22688-79-5), 5-O-caffeoylquinic acid (14534-61-3), quercetin 3-O-rutinoside (rutin) (153-18-4), gallocatechin (1617-55-6), epicatechin gallate (1257-08-5), quercetin-3-O-B-D-glucuronide (22688-79-5), syringic acid (530-57-4), luteolin-7-O-glucoside (5373-11-5), and daidzein (486-66-8).

Furthermore, other reagents, identified by their respective CAS numbers in parentheses, were obtained from Extrasynthese (Genay, France), including mCoumaric (3-coumaric acid) (614-60-8), oCoumaric (2-coumaric acid) (614-60-8), cyanidin chloride (528-58-5), petunidin chloride (1429-30-7), diosmin (520-27-4), isoferulic acid, 2-3 dihydroxybenzoic acid (537-73-5), gentisic acid (490-79-9), verbascoside (61276-17-3), formononetin (485-72-3), apigenin (520-36-5), apigenin-7-O-glucoside (578-74-5), delphinidin chloride (528-53-0), hesperetin (69097-99-0), glycitein (40957-83-3), vanillin (121-33-5), myricetin (529-44-2), chicoric acid (70831-56-0), vanillic acid (121-34-6), eriodictyol (4049-38-1), quercetin 3-O-galactoside (482-36-0), kaempferol 3-O-rutinoside (17650-84-9), kaempferol 3-O-glucoside (480-10-4), epigallocatechin (970-74-1), luteolin (491-70-3), gentisic acid (490-79-9), and syringic acid (530-57-4). All reagents employed in this study were procured from Sigma Aldrich and Extrasynthese, ensuring analytical-grade quality with a minimum purity level of ≥99% unless explicitly specified otherwise. Acetonitrile and methanol (HPLC-grade) were sourced from Labscan (Labscan Asia, Bangkok, Thailand). Table A1: Summary of the properties of compounds detected in RL through LC-MS/MS.

Raspberry Leaf Sample Collection: This investigation involved the procurement of six RL samples from diverse sources: (A) Croatia—a Starwest Botanicals product was obtained through IHerb (Irvine, CA, USA), originating from Croatia as a product of the USA; (B) Bulgaria—a Frontier Co-op product obtained through IHerb, originating from Bulgaria as a product of the USA; (C) United Kingdom—a Botanical World product obtained through Amazon (Seattle, WA, USA); (D) Belgium—a Valley of Tea product obtained through Amazon; (E) Germany—a Natural Health 4 Life product obtained through Amazon; and (F) Poland—a Spice Mart product obtained through Amazon. All samples were procured in a dried state for this study’s purposes.

Polyphenol Extraction Procedure: Polyphenols in each RL sample were extracted in triplicate, following the method outlined by [99] with certain modifications by [100]. Specifically, 0.5 mL of 80:20 (*v*/*v*) methanol/acidified water (99.90% methanol and 0.1% formic acid) was added to 50 mg of the dried RL sample (ground). Each mixture underwent 30 min of sonication at room temperature. Subsequently, the mixture of each sample was centrifuged at 14,000 rpm for 10 min, and the supernatant was collected. This process was repeated using another 0.5 mL of 80:20 *v*/*v* methanol/acidified water extract with 25 min of sonication at room temperature. The resulting mixture of each sample was centrifuged, and the supernatants were combined, filtered using a 0.22 μm pvdf syringe filter, and stored in the freezer at −18 °C until LC-MS/MS analysis.

Standard and Sample Solutions: An initial primary solution comprising all standards was prepared using methanol. This solution involved mixing 1 mg of all standards with 1 mL of 80:20 (*v*/*v*) methanol/acidified water (99.90% methanol and 0.1% formic acid). To create calibration curves covering concentrations from 0.01 to 5 µg/mL, these standardised solutions were diluted with acidified water containing 0.1% formic acid in a 1:1 *v*/*v* ratio. This process resulted in 10 calibration points. The stability of the primary stock solution was maintained for up to 3 months by storing it at −18 °C before its intended use.

Liquid Chromatography–Mass Spectrometry (LC-MS) Analysis of Raspberry Leaf: A LC-MS/MS system (LCMS-8050; Shimadzu, Kyoto, Japan) was employed for the unbiased analysis of ellagitannin, flavonoids, and phenolic acids in RL. Chromatographic separation was conducted using an ACQUITY UPLC HSS T3 column (100 mm × 2.1 mm, 1.8 μm, Waters, Milford, MA, USA) at a column temperature of 40 °C, with a flow rate of 0.5 mL/min. The retention time and mass spectrometric characteristics of phenolic compounds identified by LC-MS/MS are presented in Table A1.

Following the approach outlined in [101] (p. 2), the LC × LC instrument was coupled to an LCMS-8050 mass spectrometer using an ESI source (Shimadzu, Kyoto, Japan). The quantification of polyphenols in human bio-fluids was performed using LC-MS/MS. The analysis was conducted with the ACQUITY HSS T3 1.8 μm, 100 Å, 2.1 × 100 mm (p/n 186003539) column, equipped with a guard column to ensure optimal performance and longevity. The mobile phase consisted of two components: 0.1% formic acid in water (Mobile Phase A) and 0.1% formic acid in acetonitrile (MeCN, Mobile Phase B). The system operated under a back pressure of 710 bar to maintain a steady flow rate and efficient separation.

Data acquisition was carried out using the ASTM method for noise calculation. The parameters included a noise calculation time starting at 0.0 min and ending at 15.0 min, with a threshold set at 5000 µV. The maximum analysis time was capped at 15.0 min. An auto-purge sequence was employed to ensure the system’s readiness and reproducibility. The purge protocol included no specific purge order for the four phases, each with a purge time of 5 min. Pump A and Pump B, both LC-30AD models, were used along with rinse ports R1 and R2, and the default RO. The system activated automatically after the auto-purge sequence, with a total flow rate of 0.0000 mL/min and a warm-up wait time of 0 min.

The LC gradient programme was configured with the following time and concentration settings: at 2.00 min, the B concentration was 5%; at 12.00 min, it was 64%; at 14.50 min, it was 95%; and it returned to 5% at 17.00 min, which also marked the stop command. This gradient ensured the proper elution of polyphenols from the column within the specified run time. The LC stop time was set at 17.00 min for all acquisition times to ensure consistent data collection across all samples.

The interface used for the analysis was Electrospray Ionisation (ESI) with the following conditions: a nebulising gas flow of 3 L/min, a heating gas flow of 10 L/min, an interface temperature of 400 °C, a desolation temperature of 550 °C, a DL temperature of 150 °C, and a drying gas flow of 10 L/min. The autosampler pretreatment beginning was set to off, with no overlap time specified. The column oven temperature was maintained at 30 °C with a temperature limit of 90 °C to ensure optimal separation conditions. The readiness of the column was checked before each run to ensure system reliability and data accuracy.

The system was operated in binary gradient mode with a total flow of 0.6000 mL/min, a Pump B concentration of 5.0%, and a Pump B curve of 0. Configured pumps included Pump A and Pump B, both LC-30AD models, with the solenoid valve FCV-11ALS (Valve A: 1). The pressure limits for Pumps A and B were set with a maximum of 1200 bar and a minimum of 50 bar.

Noise calculation was performed using the ASTM method, starting at 0.0 min and ending at 15.0 min, with a threshold of 50 µV and a drift of 0.0 µV/min. The maximum analysis time was 15.0 min, and the failure action was set to skip. The autosampler model used was SIL-30AC, with a needle stroke of 52 mm and a sampling speed of 5.0 µL/s. The sample rack accommodated 1.5 mL vials, with a total of 105 vials. Rinse settings were configured for external rinsing only, with rinse mode set to before and after aspiration, a rinse dip time of 2 s, and a rinse time of 2 s. The cooler temperature for the autosampler was maintained at 7 °C.

Brewing of Raspberry Leaf Tea at Graded Times: RL tea weighing 2 g, sourced from the United Kingdom, was brewed with 200 mL of Buxton water (Nestle Waters UK Ltd., Slough, UK) at 100 °C. The same sample tea infusion was used for sampling at different time intervals. Steeping times varied at intervals of 0.5, 1, 2, 3, 4, 5, 10, 15, and 20 min, with each duration replicated thrice. The brewed tea was then cooled and analysed to quantify the presence of flavonoids, and total phenolics. This approach was chosen to track the compositional changes over time within the same infusion, ensuring consistency in extraction conditions and minimising variability between replicates.

### Method Validation

The LC-MS/MS method was validated in accordance with the approaches in [102,103,104] for Bioanalytical Method Validation, evaluating key parameters such as linearity (R^2^), accuracy (recovery %), the limit of quantification (LOQ), the limit of detection (LOD), and relative standard deviation (RSD), calculated as RSD (%) = (SD/mean) × 100. Precision and accuracy were assessed through repeated extract injections and triplicate extractions. Intra-day precision was determined by analysing three replicates of samples within a single day, while inter-day precision was evaluated by comparing three replicates of spiked samples analysed over three different days.

Statistical Analysis: Statistical Package for the Social Sciences (SPSS), version 29, from the International Business Machines Corporation (Armonk, NY, USA) was utilised for the analysis of results. Mean values ± standard deviations (SDs) of the means from triplicate analyses are reported. Statistical significance was determined at *p* < 0.05, employing one-way analysis of variance (ANOVA) followed by subsequent multiple comparisons using Tukey’s post hoc test.

## 5. Conclusions

This study employed LC-MS/MS to analyse the polyphenol content in RL tea samples from six locations, identifying and quantifying 37 compounds across different extractions. The results highlighted the significant impact of solvent choice on extraction efficiency and detection, with methanol and water yielding distinct polyphenol profiles. Notable variability was observed in antioxidant composition, with the UK sample exhibiting the highest total polyphenol content, while the samples from Bulgaria and Poland excelled in flavonoid, phenolic acid, and ellagic acid content, respectively. In contrast, the samples from Croatia and Belgium showed the lowest antioxidant levels. Optimal brewing times varied: polyphenols and ellagic acid peaked at 5 min, flavonoids at 4 min, and phenolic acids at 15 min, while prolonged brewing reduced flavonoid content. The observed compositional differences appear to be closely linked to the geographical origins of the samples. Factors such as climate, soil composition, altitude, and agricultural practices likely contribute to these variations. For instance, the high polyphenol content in the UK sample, the high flavonoid content in the Bulgarian sample, and the elevated levels of phenolic acids and ellagic acid in the Polish sample may be influenced by the unique environmental conditions and farming practices in these regions. Similarly, the lower antioxidant levels observed in the Croatian and Belgian samples might be due to differences in climate or soil composition in these regions. These findings underscore the importance of geographical origin in shaping the chemical composition of RL tea. In addition to geographical and environmental factors, the variety or cultivar of raspberry plants may also play a significant role in the composition of active compounds. Raspberry cultivars differ in characteristics such as fruit size, taste, aroma, and flowering patterns, which could influence the phytochemical profile of their leaves. Future research should investigate the potential differences in polyphenol content and antioxidant activity among leaves from different raspberry varieties to provide a more comprehensive understanding of the factors influencing RL tea’s bioactive properties. By exploring the interplay between geographical origin, environmental conditions, and cultivar-specific traits, this study lays the groundwork for further research into the optimisation of RL tea production and its potential health benefits.

## Figures and Tables

**Figure 1 molecules-30-00970-f001:**
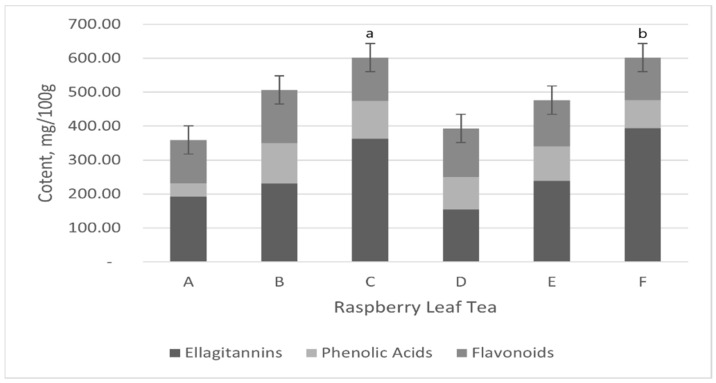
Comparative analysis of total polyphenol content in different raspberry leaf tea samples. Light grey: flavonoids; black: ellagitannins; white: phenolic acids, *p* < 0.001 (mean ± S.D., n = 3). Mean concentrations of polyphenol, flavonoids, ellagitannins, and phenolic acids in teas from different countries. Data are presented as mean ± standard deviation (n = 3). Samples include tea from Croatia (A), Bulgaria (B), the United Kingdom (C), Belgium (D), Germany (E), and Poland (F). Bars sharing the same letter indicate significant differences (*p* < 0.001 vs. all sample teas marked “a”; *p* < 0.05 vs. A, B, D, and E marked “b”).

**Figure 2 molecules-30-00970-f002:**
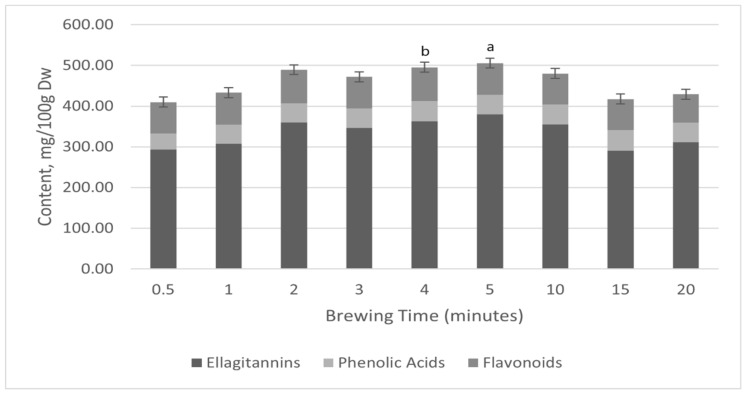
Influence of brewing time on the polyphenol content in raspberry leaf tea sample C from the UK, *p* < 0.001 (mean ± S.D., n = 3). Light grey: flavonoids; black: ellagitannins; white: phenolic acids. Mean concentrations of polyphenols, flavonoids, ellagitannins, and phenolic acids after brewing time in tea C from the UK. Data are presented as mean ± standard deviation (mean ± S.D., n = 3). Bars sharing the same letter indicate significant differences (*p* > 0.5 for 2, 3, 4, and 10 min marked “a”; *p* < 0.001 for 0.50, 15, and 20 marked “b”).

**Table 1 molecules-30-00970-t001:** Polyphenols content in raspberry leaf (mg/100 g DW), bold figures indicate the sample with the highest concentration of each individual compound.

Peak no	Rt (min)	Compound Identity	Content, mg/100 g DW ± SD					
			A	B	C	D	E	F
1	1.295	Gallic Acid	6.45 ± 0.42	8.88 ± 0.48	12.45 ± 0.31	6.43 ± 1.30	8.61 ± 0.30	**16.23 ± 0.87**
2	2.624	2,3-Dihydroxybenzoic Acid	1.40 ± 0.04	5.93 ± 0.06	**6.50 ± 0.17**	4.31 ± 0.01	4.77 ± 0.09	4.95 ± 0.15
3	3.552	Neochlorogenic Acid	0.90 ± 0.04	**3.46 ± 0.68**	0.26 ± 0.06	0.42 ± 0.10	0.41 ± 0.07	0.64 ± 0.06
4	3.993	Gentisic Acid	1.84 ± 0.06	2.56 ± 0.09	5.64 ± 0.09	1.85 ± 0.15	5.86 ± 0.04	**21.52 ± 0.04**
5	4.48	Catechin	3.19 ± 0.87	2.69 ± 0.13	2.49 ± 0.27	**3.89 ± 0.55**	1.00 ± 0.22	0.87 ± 0.06
6	4.625	Cryptochloroqenic Acid	0.91 ± 0.06	**1.56 ± 0.21**	1.49 ± 0.01	0.67 ± 0.08	1.16 ± 0.23	0.93 ± 0.05
7	4.652	Vanillic Acid	6.17 ± 0.43	3.95 ± 0.86	2.44 ± 2.73	4.31 ± 0.36	3.51 ± 2.51	**7.00 ± 0.46**
8	4.79	Caffeic Acid	2.09 ± 0.08	6.74 ± 0.29	8.22 ± 0.38	5.42 ± 0.22	**8.36 ± 0.66**	7.04 ± 0.66
9	4.964	Syringic Acid	0.38 ± 0.04	0.33 ± 0.09	0.48 ± 0.16	0.24 ± 0.02	0.38 ± 0.10	**0.66 ± 0.27**
10	5.115	Chlorogenic acid	18.22 ± 1.08	**70.96 ± 2.70**	69.01 ± 2.03	68.79 ± 2.66	64.01 ± 2.65	21.51 ± 1.02
11	5.655	P-Coumaric Aicd	0.11 ± 0.02	**1.74 ± 0.04**	1.46 ± 0.02	0.81 ± 0.09	1.04 ± 0.04	0.96 ± 0.05
12	6.01	Ferulic Acid	0.25 ± 0.01	0.96 ± 0.02	0.90 ± 0.23	**0.96 ± 0.20**	0.88 ± 0.23	0.49 ± 0.08
13	6.106	Verbascoside	ND	**0.33 ± 0.13**	ND	ND	ND	ND
14	6.126	Luteolin-7-O-Glucoside	0.87 ± 0.55	2.17 ± 1.03	1.89 ± 0.47	2.00 ± 0.55	**3.15 ± 0.66**	1.25 ± 0.46
15	6.127	Quercetin-3-Glucoside	1.85 ± 0.04	1.32 ± 0.10	2.30 ± 0.13	1.61 ± 0.14	**2.42 ± 0.24**	1.53 ± 0.07
16	6.13	Quercetin-3-O-Rutinoside	3.52 ± 0.19	9.64 ± 0.66	8.29 ± 1.44	9.33 ± 0.48	**10.11 ± 1.39**	3.92 ± 0.28
17	6.144	Mcoumaric	ND	0.14 ± 0.10	**1.90 ± 0.05**	0.34 ± 0.08	1.66 ± 0.09	0.05 ± 0.00
18	6.194	Isoferulic Acid	0.13 ± 0.01	0.27 ± 0.07	**0.35 ± 0.06**	0.26 ± 0.03	0.31 ± 0.16	0.09 ± 0.05
19	6.215	Quercetin 3-O-galactoside	1.77 ± 0.03	1.31 ± 0.09	2.25 ± 0.09	1.58 ± 0.14	**2.48 ± 0.32**	1.52 ± 0.04
20	6.457	Kaempferol-O-Rutinoside	**0.42 ± 0.12**	ND	0.07 ± 0.00	0.40 ± 0.33	ND	ND
21	6.64	Kaempferol-3-O-glucoside	0.86 ± 0.24	1.48 ± 0.92	1.66 ± 0.38	1.80 ± 0.71	**2.87 ± 0.50**	1.07 ± 0.16
22	6.554	Diosmin	0.60 ± 0.04	**14.34 ± 1.79**	0.91 ± 0.10	0.77 ± 0.04	2.03 ± 0.27	1.58 ± 0.36
23	6.632	Ocoumaric	ND	**0.27 ± 0.00**	0.03 ± 0.00	0.04 ± 0.00	0.06 ± 0.05	0.02 ± 0.00
24	6.928	Myricetin	0.12 ± 0.01	0.18 ± 0.02	0.15 ± 0.01	0.13 ± 0.02	0.13 ± 0.00	**0.26 ± 0.04**
25	6.981	Rosmarinic Acid	0.03 ± 0.01	10.95 ± 1.73	ND	**0.28 ± 0.04**	0.26 ± 0.04	ND
26	7.455	Daidzein	0.23 ± 0.00	**0.29 ± 0.00**	ND	0.23 ± 0.00	0.25 ± 0.03	ND
27	7.628	Eriodictyol	0.35 ± 0.06	0.46 ± 0.14	0.34 ± 0.06	0.16 ± 0.06	**0.77 ± 0.15**	0.56 ± 0.06
28	7.64	Glycitein	ND	**0.05 ± 0.00**	ND	ND	ND	ND
29	7.773	Luteolin	0.16 ± 0.06	0.99 ± 0.06	0.77 ± 0.03	**1.10 ± 0.19**	0.63 ± 0.12	0.15 ± 0.06
30	7.829	Quercetin	2.96 ± 0.29	9.17 ± 0.61	5.52 ± 0.60	3.25 ± 0.09	9.64 ±0.19	**11.24 ± 0.47**
31	7.899	Quercetin-3-O- glucuronide	90.67 ± 3.50	92.71 ± 3.08	77.25 ± 3.93	90.01 ± 6.73	83.87 ± 9.26	**90.91 ± 5.14**
32	8.167	Epicatechin	17.68 ± 0.22	12.18 ± 0.81	18.38 ± 1.13	**24.11 ± 4.63**	9.56 ± 1.10	4.64 ± 0.87
33	8.443	Naringenin	0.27 ± 0.03	0.48 ± 0.08	0.45 ± 0.03	0.26 ± 0.03	0.67 ± 0.11	**0.71 ± 0.10**
34	8.683	Kaempferol	0.87 ± 0.12	3.92 ± 0.15	2.20 ± 0.21	1.51 ± 0.20	**4.00 ± 0.25**	3.47 ± 0.18
35	8.718	Hesperetin	0.20 ± 0.01	0.19 ± 0.02	0.21 ± 0.02	0.19 ± 0.01	0.20 ± 0.01	**0.21 ± 0.03**
36	8.795	Isoquercitrin	0.90 ± 0.47	**3.15 ± 0.54**	1.36 ± 0.43	0.83 ± 0.19	2.10 ± 0.89	1.16 ± 0.21
37	9.586	Ellagic Aicd	192.32 ± 20.72	230.84 ± 14.46	363.36 ± 54.67	155.27 ± 11.11	238.9 ± 12.2	**394.22 ± 21.98**
Total of Ellagitannin content (mg/100 g)			192.32 ± 20.72	230.84 ± 14.46	363.36 ± 54.67	155.27 ± 11.11	238.97 ± 12.27	**394.22 ± 21.98 ^#^**
Total of Flavonoid content (mg/100 g)			127.47 ± 4.16	**156.73 ± 5.02 ****	127.16 ± 1.86	143.16 ± 3.09	135.87 ± 10.56	125.03 ± 6.18
Total of Phenolic Acid content (mg/100 g)			38.87 ± 1.38	**119.03 ± 3.89 ^&^**	111.13 ± 4.49	95.12 ± 3.51	101.28 ± 1.48	82.12 ± 1.79
Total of Polyphenol content (mg/100 g)			358.66 ± 16.90	506.60 ± 13.91	**601.65 ± 51.59 ***	393.54 ± 9.09	476.13 ± 3.2	601.38 ± 16.59 *

Analysis of teas from different countries: Croatia (A), Bulgaria (B), the United Kingdom (C), Belgium (D), Germany (E), and Poland (F). Data include peak number (Peak no.), retention time (Rt), and compound concentrations (ND: not detected). Values are expressed as mean ± S.D. of three determinations (n = 3) and are reported in mg/100 g. Results sharing the same symbols indicate significant differences *p* < 0.001 (Total Polyphenol: teas C and F vs. teas A, B, D, and E, marked “*”; Ellagitannins: tea F vs. teas A, B, D, and E, marked “#”; Flavonoids: tea B vs. A, marked “**”; Phenolic Acids: tea B vs. teas A, D, E, and F, marked “&”).

**Table 2 molecules-30-00970-t002:** Polyphenols content in raspberry leaf tea, sample C, UK, (mg/100g DW), bold figures indicate the sample with the highest concentration of each individual compound.

Peak no	Rt (min)	Compound Identity	Brewing Time (Minutes)								
			0.5	1	2	3	4	5	10	15	20
1	1.295	Gallic Acid	4.35 ± 0.35	4.74 ± 0.31	5.71 ± 0.17	6.19 ± 0.30	6.43 ± 0.41	6.78 ± 0.18	7.46 ± 0.27	7.74 ± 0.30	**8.09 ± 0.14**
2	2.624	2,3-Dihydroxybenzoic Acid	1.53 ± 0.06	1.60 ± 0.04	1.74 ± 0.03	1.73 ± 0.08	1.71 ± 0.06	1.78 ± 0.03	**1.84 ± 0.07**	1.82 ± 0.03	1.78 ± 0.07
3	2.923	Gallocatechin	ND	**0.21 ± 0.00**	ND	ND	**0.21 ± 0.00**	ND	ND	ND	ND
4	3.552	Neochlorogenic Acid	**0.76 ± 0.93**	0.33 ± 0.03	0.34 ± 0.01	0.36 ± 0.00	0.37 ± 0.08	0.46 ± 0.01	0.51 ± 0.03	0.50 ± 0.03	0.53 ± 0.10
5	3.993	Gentisic Acid	2.37 ± 0.04	2.54 ± 0.06	**2.61 ± 0.01**	2.40 ± 0.14	2.51 ± 0.08	2.49 ± 0.09	2.36 ± 0.19	2.39 ± 0.13	2.50 ± 0.12
6	4.244	Epigallocatechin	**0.21 ± 0.00**	0.20 ± 0.00	0.20 ± 0.00	0.20 ± 0.00	0.20 ± 0.00	0.20 ± 0.00	0.20 ± 0.00	0.20 ± 0.00	0.20 ± 0.00
7	4.445	Chlorogenic Acid	14.13 ± 10.73	21.44 ± 0.94	21.41 ± 1.47	21.50 ± 1.02	21.77 ± 0.56	**22.07 ± 0.88**	21.03 ± 0.94	21.63 ± 0.56	19.67 ± 1.90
8	4.48	Catechin	0.36 ± 0.06	0.44 ± 0.04	0.52 ± 0.02	0.53 ± 0.07	**0.64 ± 0.04**	0.61 ± 0.07	0.63 ± 0.02	0.55 ± 0.01	0.59 ± 0.02
9	4.625	Cryptochloroqenic Acid	1.05 ± 0.69	0.66 ± 0.08	0.92 ± 0.06	0.88 ± 0.08	0.88 ± 0.06	0.92 ± 0.07	1.00 ± 0.09	**1.08 ± 0.05**	0.99 ± 0.14
10	4.652	Vanillic Acid	1.43 ± 0.14	1.42 ± 0.13	1.30 ± 0.14	1.57 ± 0.09	1.65 ± 0.26	1.47 ± 0.21	1.60 ± 0.19	**1.71 ± 0.09**	1.61 ± 0.15
11	4.688	Salicylic Acid	**0.30 ± 0.04**	ND	ND	ND	ND	ND	ND	ND	ND
12	4.79	Caffeic Acid	**3.79 ± 022**	3.66 ± 0.08	3.67 ± 0.12	3.46 ± 0.01	3.55 ± 0.17	3.53 ± 0.15	3.44 ± 0.11	3.36 ± 0.05	3.47 ± 0.18
13	4.964	Syringic Acid	ND	0.35 ± 0.06	0.31 ± 0.06	**0.37 ± 0.08**	0.29 ± 0.04	0.33 ± 0.08	0.34 ± 0.04	0.32 ± 0.05	0.31 ± 0.09
14	5.167	Epicatechin	2.00 ± 0.36	2.45 ± 0.19	3.45 ± 0.31	3.39 ± 0.27	**3.75 ± 0.45**	3.59 ± 0.05	3.70 ± 0.16	3.25 ± 0.21	3.33 ± 0.56
15	5.43	Vanillin	0.70 ± 0.03	0.57 ± 0.02	0.50 ± 0.10	0.88 ± 0.05	0.89 ± 0.08	0.61 ± 0.02	0.92 ± 0.06	**1.01 ± 0.09**	0.91 ± 0.14
16	5.655	P-Coumaric Aicd	0.92 ± 0.06	0.19 ± 0.06	0.87 ± 0.02	0.89 ± 0.03	0.95 ± 0.06	0.93 ± 0.03	0.92 ± 0.12	**0.97 ± 0.05**	0.88 ± 0.07
17	5.688	Cyanidin Chloride	3.31 ± 0.16	3.66 ± 0.06	3.25 ± 0.37	3.56 ± 0.57	**3.72 ± 0.48**	3.39 ± 0.50	3.57 ± 0.16	3.27 ± 0.25	3.18 ± 0.23
18	5.778	Petunidin Chloride	0.49 ± 0.47	0.08 ± 0.00	0.20 ± 0.04	0.03 ± 0.00	0.31 ± 0.20	0.15 ± 0.00	0.31 ± 0.00	0.36 ± 0.02	**0.51 ± 0.00**
19	6.01	Ferulic Acid	0.39 ± 0.04	0.42 ± 0.03	0.37 ± 0.02	0.41 ± 0.08	**0.44 ± 0.06**	0.38 ± 0.02	0.44 ± 0.05	0.42 ± 0.01	0.43 ± 0.04
20	6.086	Ellagic Acid	293.46 ± 16.66	307.88 ± 11.70	360.52 ± 27.78	346.50 ± 18.54	363.46 ± 19.58	**380.29 ± 35.85**	355.00 ± 19.97	290.55 ± 22.58	311.95 ± 3.25
21	6.088	Epicatechin gallate	1.52 ± 0.60	1.78 ± 0.38	1.58 ± 0.68	1.90 ± 0.84	1.78 ± 0.34	1.78 ± 1.35	**2.24 ± 1.47**	2.04 ± 0.11	1.26 ± 1.13
22	6.106	Verbascoside	**0.77 ± 0.92**	0.23 ± 0.01	0.24 ± 0.01	0.24 ± 0.01	0.24 ± 0.00	0.24 ± 0.00	0.24 ± 0.01	0.24 ± 0.00	0.24 ± 0.00
23	6.126	Luteolin-7-O-glucoside	0.78 ± 0.50	1.05 ± 0.11	0.97 ± 0.36	**1.11 ± 0.21**	0.81 ± 0.07	0.90 ± 0.07	0.91 ± 0.07	0.61 ± 0.10	0.51 ± 0.08
24	6.127	Quercetin-3-glucoside	1.54 ± 0.18	1.66 ± 0.08	1.45 ± 0.13	1.47 ± 0.17	1.52 ± 0.17	1.41 ± 0.01	1.43 ± 0.03	1.31 ± 0.06	1.31 ± 0.03
25	6.13	Quercetin-3-O-rutinoside	2.48 ± 0.23	2.40 ± 0.08	**2.82 ± 0.39**	2.64 ± 0.31	2.50 ± 0.20	2.40 ± 0.15	2.52 ± 0.30	2.27 ± 0.25	2.55 ± 0.29
26	6.144	Mcoumaric	0.75 ± 0.13	0.84 ± 0.02	0.74 ± 0.16	0.78 ± 0.12	**0.93 ± 0.06**	0.74 ± 0.02	0.76 ± 0.10	0.70 ± 0.09	0.83 ± 0.04
27	6.194	Isoferulic Acid	0.30 ± 0.02	0.27 ± 0.01	0.25 ± 0.04	0.25 ± 0.05	0.29 ± 0.07	0.26 ± 0.02	0.29 ± 0.09	0.30 ± 0.02	**0.31 ± 0.03**
28	6.215	Quercetin 3-O-galactoside	1.46 ± 0.11	**1.62 ± 0.10**	1.47 ± 0.11	1.46 ± 0.19	1.50 ± 0.18	1.42 ± 0.02	1.39 ± 0.05	1.33 ± 0.06	1.25 ± 0.03
29	6.299	Quercetin-3-O-glucuronide	65.43 ± 5.91	67.02 ± 2.84	**69.64 ± 0.94**	65.14 ± 3.76	63.73 ± 4.28	64.67 ± 4.34	63.31 ± 4.95	63.82 ± 3.08	58.23 ± 0.41
30	6.457	Kaempferol-O-rutinoside	0.29 ± 0.10	0.25 ± 0.14	0.24 ± 0.12	0.25 ± 0.06	0.14 ± 0.12	0.24 ± 0.01	0.18 ± 0.15	0.27 ± 0.01	**0.32 ± 0.20**
31	6.632	Ocoumaric	0.14 ± 0.00	0.12 ± 0.00	**0.14 ± 0.02**	0.13 ± 0.01	0.12 ± 0.00	0.12 ± 0.01	0.13 ± 0.02	0.13 ± 0.02	0.13 ± 0.02
32	6.64	Kaempferol-3-O-glucoside	0.60 ± 0.10	0.63 ± 0.04	0.64 ± 0.03	0.63 ± 0.11	0.61 ± 0.09	**0.76 ± 0.13**	0.58 ± 0.05	0.47 ± 0.09	0.42 ± 0.04
33	6.882	Phloridizin	0.49 ± 0.47	1.27 ± 0.28	0.86 ± 0.64	0.84 ± 0.59	1.45 ± 0.84	0.27 ± 0.46	0.52 ± 0.30	**2.41 ± 1.18**	0.53 ± 0.29
34	6.928	Myricetin	**0.07 ± 0.00**	0.05 ± 0.00	0.06 ± 0.00	0.06 ± 0.00	0.06 ± 0.00	0.06 ± 0.00	0.05 ± 0.00	0.05 ± 0.00	0.06 ± 0.00
35	7.829	Quercetin	0.04 ± 0.01	**0.05 ± 0.00**	0.04 ± 0.01	0.05 ± 0.01	0.02 ± 0.02	0.03 ± 0.01	0.01 ± 0.01	ND	ND
36	8.443	Naringenin	**0.74 ± 0.95**	0.20 ± 0.00	0.20 ± 0.01	0.20 ± 0.00	0.20 ± 0.00	0.19 ± 0.00	0.19 ± 0.00	0.19 ± 0.00	0.19 ± 0.00
37	8.718	Hesperetin	**0.75 ± 0.95**	0.19 ± 0.00	0.19 ± 0.00	0.20 ± 0.00	0.20 ± 0.00	0.19 ± 0.00	0.19 ± 0.00	0.19 ± 0.00	0.19 ± 0.00
Total of Ellagitannin content (mg/100 g)			293.46 ± 16.66	307.88 ± 11.70	360.52 ± 27.78	346.50 ± 18.54	364.46 ± 19.58	**380.29 ± 35.85 ^#^**	355.00 ± 19.97	29.55 ± 22.58	311.95 ± 3.20
Total of Flavonoid content (mg/100 g)			77.47 ± 7.34	79.16 ± 2.72	82.51 ± 0.22	78.11 ± 3.36	**82.58 ± 4.88 ****	77.56 ± 4.88	76.62 ± 4.31	75.94 ± 2.99	69.89 ± 1.56
Total of Phenolic Acid content (mg/100 g)			38.90 ± 7.13	46.14 ± 0.98	46.40 ± 1.91	47.57 ± 1.60	49.30 ± 0.70	47.80 ± 1.50	48.60 ± 1.22	**50.96 ± 0.72 ^&^**	47.43 ± 1.33
Total of Polyphenol content (mg/100 g)			409.84 ± 13.68	433.18 ± 14.69	489.43 ± 27.66	472.18 ± 20.29	489.80 ± 20.88	**505.65 ± 30.03 ***	480.21 ± 14.95	417.45 ± 19.41	429.27 ± 1.90

Peak number (Peak no.), retention time (Rt), and compound concentrations in teas (ND: not detected). Values are expressed as mean ± S.D. of three determinations (n = 3) and are reported in mg/100 g. Results sharing the same symbols indicate significant differences *p* < 0.001 (Total Polyphenol: 5 min vs. 0.50, 1, 15, and 20 min, marked “*”; Ellagitannins: 5 min vs. 0.50, 1, 15, and 20 min, marked “#”; Flavonoids: 4 min vs. 20 min marked “**”; Phenolic Acids: 15 min vs. 0.50 min, marked “&”).

## Data Availability

The original contributions presented in this study are included in the article. Further inquiries can be directed to the corresponding author.

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
