# Peer review of "Determination of the Polyphenol Composition of Raspberry Leaf Using LC-MS/MS"

_molecules, 2025, doi:10.3390/molecules30040970_

Round 1

Reviewer 1 Report

Comments and Suggestions for Authors

The author has investigated the polyphenol content in Raspberry leaf (RL) samples from different origins and the differences in polyphenol content due to various infusion times. The topic of this study is somewhat appealing and worth exploring. However, the manuscript still requires further revision by the author to meet the publication standards of the journal.

1、In Table 1, the size of the significance symbols “*” for C and F appears to be inconsistent.

2、Based on Table 1 and Figure 1, the author mentions, “Detailed data are presented in Table 1 and Figure 1, illustrating these compositional differences and their significance. These results highlight the significant geographical and compositional variation in RL teas.” However, in the conclusion, the author does not further elaborate on the connection between the sample results and the origins. Can the geographical differences of the six samples be briefly explained? Are these geographical differences responsible for the compositional and content variations? The reviewer believes that the author should link the samples to their origins, as this would make the conclusion more meaningful.

3、The author should explain why Sample C was chosen for the hot water infusion experiment. Does it have some representativeness? If the purpose is to observe the differences between hot water extraction and ethanol extraction, it might be more valuable to select samples with the highest and lowest polyphenol content to investigate the impact of these two extraction methods on the components and content.

4、Regarding the results in Figure 3, the author mentions in the experimental methods that the tea leaves were infused in hot water for different time intervals, followed by sampling and analysis. It can be understood that the same replicate experiment was conducted by sampling at different times from the same cup of tea. The author found that the polyphenol content decreased after 15 minutes. The polyphenols that decreased in the cup of tea between 5 and 15 minutes might have undergone various reactions, including oxidation. The reviewer believes that the author should provide a more comprehensive explanation for this part.

5、The reviewer suggests that the author should integrate the discussion more closely with the experimental results and further refine the content of the Discussion section.

6、Figures and tables are not standardized and need to be further optimized by the authors

Author Response

We were delighted to be given the opportunity to improve our manuscript in order to make it suitable for publication in the Molecules Journal, and we thank the reviewers for their valuable comments and expertise.  Please find attached a revised manuscript in which we have added novel data.  We have also responded to all of the comments of the reviewers and have made significant alterations to the manuscript, which appear in red text for clarity.  Kindly find the specific requested revisions outlined below:

Editorial report:

We thank the reviewer for their positive comments and for the suggestions given. We believe that the changes made, greatly improve the strength of the manuscript.

  1. “In Table 1, the size of the significance symbols “*” for C and F appears to be inconsistent”

Thank you for your careful observation. We have now ensured that the size of the significance symbols (“*”) for C and F in Table 1 is consistent throughout the table. This correction has been made in the revised manuscript.

  1. “Based on Table 1 and Figure 1, the author mentions, “Detailed data are presented in Table 1 and Figure 1, illustrating these compositional differences and their significance. These results highlight the significant geographical and compositional variation in RL teas.” However, in the conclusion, the author does not further elaborate on the connection between the sample results and the origins. Can the geographical differences of the six samples be briefly explained? Are these geographical differences responsible for the compositional and content variations? The reviewer believes that the author should link the samples to their origins, as this would make the conclusion more meaningful”

Thank you for your valuable feedback. We appreciate your suggestion to elaborate further on the geographical differences between the six samples and their connection to compositional variations. We have revised the conclusion to explicitly discuss the relationship between the sample origins and their compositional differences. The geographical variations among the six samples are influenced by climate, soil composition, altitude, and regional agricultural practices, contributing to differences in chemical composition and bioactive content. These environmental factors play a crucial role in shaping the unique characteristics of RL teas from different regions.  To address this point more clearly, we have added a brief explanation linking the compositional variations to their respective geographical origins in the conclusion section. We believe this revision strengthens the study’s findings and enhances the overall coherence of the discussion. 

  1. “The author should explain why Sample C was chosen for the hot water infusion experiment. Does it have some representativeness? If the purpose is to observe the differences between hot water extraction and ethanol extraction, it might be more valuable to select samples with the highest and lowest polyphenol content to investigate the impact of these two extraction methods on the components and content.”

Thank you for your valuable feedback. We would like to clarify that the goal of our study was not to compare the effects of extraction using hot water versus methanol. Instead, we initially used methanol extraction to obtain the highest possible number of polyphenols and to identify the sample with the highest polyphenol content. This approach was chosen because certain compounds are more soluble in methanol than in water. Subsequently, hot water infusion was used for further analysis because water is the most relevant solvent for human consumption. The effect of steeping time was then examined based on this selection. We assumed that all samples would respond similarly in the brewing experiment. Therefore, we selected the one with the highest total content for our analysis. This rationale has already been outlined in the methodology, results, and discussion sections, but we will further emphasize it to ensure clarity.

  1. “Regarding the results in Figure 3, the author mentions in the experimental methods that the tea leaves were infused in hot water for different time intervals, followed by sampling and analysis. It can be understood that the same replicate experiment was conducted by sampling at different times from the same cup of tea. The author found that the polyphenol content decreased after 15 minutes. The polyphenols that decreased in the cup of tea between 5 and 15 minutes might have undergone various reactions, including oxidation. The reviewer believes that the author should provide a more comprehensive explanation for this part”.

Thank you for your insightful comment. Yes, the same sample of tea infusion was used for sampling at different time intervals, as described in the experimental methods. This approach was chosen to track the compositional changes over time within the same infusion, ensuring consistency in extraction conditions and minimizing variability between replicates. We appreciate your observation regarding the potential reactions affecting polyphenol content during infusion.  To address this, we have expanded the discussion related to Figure 3 to provide a more comprehensive explanation of the observed decrease in polyphenol content after 15 minutes. As we suggested, polyphenols may undergo oxidation, hydrolysis, or polymerization over time, particularly under prolonged exposure to heat and oxygen. Additionally, interactions with other compounds, such as proteins and metal ions present in the tea, could contribute to changes in polyphenol stability.  We have now incorporated these possible mechanisms into the revised manuscript to provide a clearer interpretation of the results. Thank you for your valuable feedback, which has helped improve the clarity and scientific rigour of our discussion. 

  1. “The reviewer suggests that the author should integrate the discussion more closely with the experimental results and further refine the content of the Discussion section”.

Thank you for your valuable suggestion. We have carefully revised the Discussion section to ensure a closer integration with the experimental results. Specifically, we have refined the content by directly linking key findings to relevant literature, providing more in-depth explanations of the observed trends, and emphasizing their implications.  We appreciate your constructive feedback and believe that the improved Discussion section now provides a more comprehensive interpretation of our results.

  1. “Figures and tables are not standardized and need to be further optimized by the authors”.

Thank you for your feedback. We have carefully reviewed and standardized all figures and tables to ensure consistency in formatting, labelling, and presentation.

  1. “Introduction: Based on the literature evidence, could the Authors please discuss the issues (including potential risks) related to BL consumption during pregnancy. The evidence for BL's activity during gestation is, to the best of my knowledge, somewhat inconclusive, but it would interesting to present this topic in detail.”

Thank you for your valuable suggestion. We have now included a detailed discussion on the potential risks and issues related to RL consumption during pregnancy, based on the available literature. While the evidence on RL's activity during gestation remains inconclusive, we have presented the relevant studies and highlighted the gaps in current research. This addition can be found in red in the Introduction section of the revised manuscript. 

  1. “Why is "extrasynthese" written in lowercase? Is this a company name?”

Thank you for your observation. Extrasynthese is indeed a company name. We have now corrected the formatting to ensure proper capitalization and alignment with standard conventions. This correction has been made in the revised manuscript. 

  1. “Table 1 - please highlight the sample with the highest concentration of the particular compounds”

Thank you for your suggestion. We have now highlighted the samples with the highest concentration of each particular compound in Table 1 and 2 to enhance clarity and facilitate comparison. This update is reflected in the revised manuscript.

  1. “I couldn't find any details of calibration curves and their validation (e.g. precision, accuracy, LOD) - please add (e.g. in Supplementary Materials at the end of the text or in a separate file).”

Thank you for your valuable suggestion. We have now included the details of the calibration curves and their validation (including precision, accuracy, and LOD) in the Supplementary Materials (Appendix A-Table 1).

  1. The Authors mentioned the importance of region and harvesting time for the concentrations of particular actives - this is, of course, true, but there is also an additional factor that may/may not influence the contents of actives in RL - variety/cultivar. Raspberry fruit from different cultivars differ significantly in size, taste and aroma; some varieties fruit only once a year, the others blossom repeatedly etc. Are there any plans to investigate the differences (if any) between the composition of leaves from different varieties as well?”

Thank you for your insightful comments and for highlighting the importance of raspberry cultivar differences in influencing the phytochemical profile of RL leaves. We appreciate your suggestion and agree that genetic variation among raspberry cultivars could significantly impact the concentrations of bioactive compounds. While our current study primarily focuses on the influence of region, we recognize that cultivar differences are an important factor worth exploring. We have now acknowledged this in the discussion section and suggested it as a potential direction for future research. Additionally, we are considering further studies involving comparative phytochemical analysis and metabolomic profiling across different raspberry cultivars to better understand these variations. We appreciate your valuable input and have incorporated this perspective into the revised manuscript.

  1. “There are several minor, yet annoying errors in the text (missing prepositions etc.).”

Thank you for your feedback. I have conducted a proofreading review through the services provided by the journal and have made the necessary linguistic improvements. I appreciate your keen attention to detail and efforts to ensure high-quality content.

Again, we express our thanks to the reviewers and we hope that our answers to their comments are satisfactory.

We look forward to hearing from you.

Yours Sincerely

Hind Alkhudaydi

Reviewer 2 Report

Comments and Suggestions for Authors

The topic of this manuscript is relevant and sufficiently interesting to justify publications after the following issues are addressed:

1. Introduction: Based on the literature evidence, could the Authors please discuss the issues (including potential risks) related to BL consumption during pregnancy. The evidence for BL's activity  during gestation is, to the best of my knowledge, somewhat inconclusive, but it would interesting to present this topic in detail.

2. Why is "extrasynthese" written in lowercase? Is this a company name?

3. Table 1 - please highlight the sample with the highest concentration of the particular compounds

4. I couldn't find any details of calibration curves and their validation (e.g. precision, accuracy, LOD) - please add (e.g. in Supplementary Materials at the end of the text or in a separate file). 

The Authors mentioned the importance of region and harvesting time for the concentrations of particular actives - this is, of course, true, but there is also an additional factor that may/may not influence the contents of actives in RL - variety/cultivar.   Raspberry fruit from different cultivars differ significantly in size, taste and aroma; some varieties fruit only once a year, the others blossom repeatedly etc. Are there any plans to investigate the differences (if any) between the composition of leaves from different varieties as well?

Comments on the Quality of English Language

There are several minor, yet annoying errors in the text (missing prepositions etc.).

Author Response

(The authors gave the same response as above.)

Round 2

Reviewer 1 Report

Comments and Suggestions for Authors

The author has made substantial revisions to the previous issues, and the reviewer concurs that the article can be published following further refinement.

Reviewer 2 Report

Comments and Suggestions for Authors

I am satisfied with the corrections and the Authors' replies to my comments.